# Abortion in Countries with Restrictive Abortion Laws—Possible Directions and Solutions from the Perspective of Poland

**DOI:** 10.3390/healthcare9111594

**Published:** 2021-11-20

**Authors:** Kornelia Zaręba, Krzysztof Herman, Ewelina Kołb-Sielecka, Grzegorz Jakiel

**Affiliations:** First Department of Obstetrics and Gynecology, Center of Postgraduate Medical Education, 01-004 Warsaw, Poland; paniehermanie@gmail.com (K.H.); ewelina.sielecka@gmail.com (E.K.-S.); grzegorz.jakiel1@o2.pl (G.J.)

**Keywords:** abortion, pregnancy termination, abortion law, abortion history, TOP

## Abstract

The tendency towards the radicalization of abortion law is observed in numerous countries, including Poland. The aim of the present paper was to determine the main factors influencing the number of abortions performed worldwide and to indicate the main directions which should be followed to improve the patients’ well-being. The authors conducted their search in the PubMed of the National Library of Medicine and Google Scholar. Databases were extensively searched for all original and review articles/book chapters in English until June 2021. The main problems associated with the contemporary policy of birth regulation include no possibility of undergoing a termination because of the conscience clause invoked by the medical personnel, restrictive abortion law and lack of sexual education. Minimal changes that should be considered are: improved sex education and the availability of contraception, free access to abortion-inducing drugs with adequate information provided by qualified medical personnel in countries with a conscience clause invoked by the personnel, and the development of an international network which would facilitate undergoing a pregnancy termination abroad to provide women with access to legal abortion assisted by professional medical personnel.

## 1. Introduction

The Conference on Population and Development held in 1994 in Cairo harmonized population policy all over the world, recognizing the women’s right of access to abortion, sterilization, and contraception. The objectives of the Cairo Action Plan permitted the adjustment of its provisions to the national laws of individual countries to respect their religious, cultural, and ethical separateness [1,2]. Regrettably, a tendency towards the radicalization of abortion law is observed in radical countries, including Poland [3,4,5]. The process is undergoing gradual progress in Poland. Currently, pregnancy termination due to severe and irreversible fetal defects is forbidden. Terminations may only be performed in the case of a threat to maternal life and if the pregnancy resulted from rape. In other countries abortion has been banned entirely [3,5].

The percentages of pregnancy terminations vary significantly in different locations worldwide [6,7]. The aim of the present review was to determine the main factors influencing the number of abortions in terms of the law determined by the culture, religion, and the accessibility of professional medical care. The majority of publications focused on the global issue of abortion and the areas such as the possible methods of performing a termination, increasing its accessibility and cost-effectiveness [8,9]. We try to indicate the main directions which should be followed to improve the patients’ well-being in a particular group of individuals in the countries with restrictive abortion laws where the main problem is linked to the fact that abortion cannot be performed.

Pregnancy termination is completely prohibited in 3 European Union countries and over 20 countries worldwide [10]. According to Berer, the safety of pregnancy termination is guaranteed by its easy accessibility [11]. According to the literature abortion law was introduced in order to limit the number of deaths in women undergoing illegal terminations, authorize socio-moral norms and difficulties related to the difficult access to pregnancy termination, and to protect human life at conception [11,12]. Pregnancy termination is completely prohibited in the many countries [10].

Currently, due to the ruling of the Polish Constitutional Tribunal dated from 22 October 2020 in which the Tribunal adjudicated that the regulation allowing pregnancy termination in the case of a high probability of severe and irreversible fetal defect was unconstitutional, pregnancy termination is only possible in the case of a threat to maternal life and rape [4]. The Tribunal declared terminations due to severe and incurable fetal defects to be incompatible with the Polish Constitution. At the same time sex education was removed from school educational programs. In November 2021, Poland faced numerous social protests associated with a death of a patient at 21 weeks of gestation. The woman died as a result of septic shock which was probably due to premature rupture of membranes. The patient was refused a termination because the fetus was still alive [13].

The main problems associated with the contemporary policy of birth regulation include no possibility of undergoing a termination because of the conscience clause invoked by the medical personnel [14]. The phenomenon was widely observed in Poland when terminations were still possible due to severe and irreversible fetal defects [15]. Similarly, over 75% of gynecologists invoke the conscience clause in the case of pregnancy termination in Italy [16]. According to Bhakuni the same reasons which determine the permissibility of abortion also provide justification for the right to refuse a termination by the medical personnel. The concept of “correspondence argument” based on the correspondence thesis which justifies co-dependent actions, confirms that invoking the conscience clause ceteris paribus is morally wrong. The “correspondence argument” runs as follows: “Preventing, obstructing, or adding a burden on morally justified termination of pregnancy is morally wrong. Conscientiously objecting to termination of pregnancy often prevents, obstructs, or adds burdens on morally justified termination of pregnancy. Therefore, conscientiously objecting to morally justified termination of pregnancy is ceteris paribus morally wrong”. Hence, it is claimed to be wrong only in the case in which it prevents, obstructs, or adds burdens to the termination of pregnancy [14].

According to Berer “the road to law reform is long and difficult… The biggest challenge is to determine what is possible to achieve, build a critical mass of support, and work together with legal experts, parliamentarians, health professionals, and women themselves to change the law” [11]. Abortion law undergoes constant changes. Numerous papers describing abortion laws worldwide were published over 20 years ago [17,18]. The present paper was prepared owing to dynamic changes in abortion law in Poland. The changes are still the subject of numerous social debates and further attempts at radicalization. Due to the inability to perform a abortion because of social and fetal indications in Poland and basing on available research results, we decided to seek possible alternatives which would enhance the situation of women facing restrictive abortion laws.

## 2. Materials and Methods

We reviewed articles concerning abortion from various countries. The authors conducted their search in the PubMed of the National Library of Medicine and Google Scholar. Databases were extensively searched for all original and review articles/book chapters using keywords (one or in combinations): abortion, pregnancy termination, termination of pregnancy, published in English until September 2021. Moreover, additional articles in the references of the reviewed articles were searched. Overall, the most relevant articles were reviewed and included as appropriate.

## 3. Results

### 3.1. Legal Aspects of Abortion in Restrictive Abortion Countries

In the majority of the European Union states, abortion is available on demand in the first trimester [19]. In high-income countries over 90% of terminations are performed before 13 weeks of gestation [19,20]. It also increases the safety of such procedures. Terminations performed later are usually due to structural fetal defects which develop at later stages of pregnancy [21]. In the second trimester, abortion may usually be performed for medical, social, or legal reasons [22]. The majority of countries permit termination on demand or due to social indications which may be explained by no wish to have a child, difficult living conditions, and possible mental disorders which might occur after childbirth [19].

Numerous countries still have restrictive abortion laws (Table 1) [23]. A complete ban on terminations was mainly approved by the countries of Africa, South America, Asian islands, and Oceania. The list of countries permitting pregnancy terminations only due to the threat to maternal health is very long and, by analogy, mostly covers African and South American countries.

Ireland was a country with the most radical abortion law in Europe. However, in May 2018, the abortion law was eased after a referendum and terminations due to medical indications were permitted [24,25]. Abortion is still prohibited in Malta [26].

In Australia, the most extensive list of indications for termination is in place in Victoria, where the main reason for termination is a threat to the broadly defined mother’s health. The northern and southern territories permit abortion until week 14 of pregnancy, while in Tasmania abortion is illegal [27]. Only one clinic performs such procedures in the state of Victoria. In terms of the availability of termination, the moral and legal situation in Australia is similar to the situation in Poland. In some regions the abortion law is restrictive, and in others—more liberal. The availability of an abortion in the country mainly depends on the place of the patient’s residence.

In MENA countries (the Middle East and North Africa), one in ten pregnancies ends in an abortion. An individual interpretation of the law is the key principle in place in Islamic countries. Muslims determine their principles of conduct based on historic Arabic writings. In Sunni Islam, there are four official schools of interpretation, with Shiites having their separate law. Sharia law differs in individual countries. In the event of unclear answers in their holy books, spiritual guides issue a fatwa (religious edict with guidance) [28]. Most scholars accept abortion, depending on the school, performed within 40, 90, or 120 days following conception. Abortion is prohibited at later stages of pregnancy–except when it is required to save the woman’s life—but only until the fetus achieves the status of a person. All the schools prohibit termination in out-of-marriage relationships. A fatwa from 1991 issued in Saudi Arabia permits termination within 120 days in the case of fetal defects. A similar law is in place in Qatar, which permits termination in the first trimester due to genetic defects and a threat to the mother’s life. Rape is also an indication for termination in Islamic countries, but women in Kuwait raped by Iraqi soldiers during the first war in the Persian Gulf were not granted the right to abortion. The mufti claimed that innocent life should be protected even if it was conceived as a result of rape [29]. Since 2005 abortion has been legal in Iran in cases when the mother’s life is in danger and in cases of fetal abnormalities before week 19 of the pregnancy [30,31].

In Nigeria, pregnancy terminations are accepted only in the case of a threat to maternal life. However, abortion is also acceptable if the patient self-administered abortion drugs [32]. Despite lots of controversies, Trinidad and Tobago permit abortion intending to save the woman’s life, including her mental health [33]. According to the literature, the number of abortions in the country is equal to the number of births at 19 thousand annually. Every year, 3–4 thousand women are treated due to the complications of illegal abortions. In a survey conducted in that country, 74% of Catholic females were in favor of the liberalization of the abortion law [33]. A study conducted in Zambia demonstrated that despite liberal abortion law the percentage of complications of unsafe abortions was still high and resulted from the lack of access to qualified medical personnel. The costs of the treatment of complications are higher by 27% than in the case of safe abortions [34].

In China, abortion has been a form of birth control and a family planning tool since 1988 [35]. Due to the decreasing birth rate a two-child policy introduced in 2015 was changed into a three-child policy in 2021.

In India it had become such a popular method of sex selection that ultrasound scans performed to determine sex were officially prohibited. Despite the ban on sex selection in India, this phenomenon still occurs, even in affluent families. Selection takes place both directly—through abortion, and indirectly—by providing better nutrition to male babies which translates into a higher survival rate. Due to sex selection, the number of females has been dropping year by year in India and China as compared to the number of males. In 1994, the Indian government officially prohibited sex selection, which was legally authorized in 2002 [35]. Due to the phenomenon becoming more and more widespread, pre-implantation diagnostic methods were introduced in 2001 for ‘family balancing’. The Ethics Committee of the ASRM recognized sex selection before implantation as a more humanitarian option [36].

### 3.2. Abortion in Poland

Compared to other European countries, the Polish law is one of the most restrictive ones in terms of indications for abortion. In Poland, similarly to other ex-Soviet Union states, legal abortion had been widely available since 1956. An act issued in 1993 set out significantly stricter indications for termination, restricting them to strict medical grounds and sexual offenses [37]. Currently, due to the ruling of the Polish Constitutional Tribunal dated from 22 October 2020 in which the Tribunal adjudicated that the regulation allowing pregnancy termination in the case of a high probability of severe and irreversible fetal defect was unconstitutional, pregnancy termination is only possible in the case of a threat to maternal life and rape [4].

According to official data, 1057 legal terminations were recorded in Poland in 2017, including 22 procedures performed due to a threat to the mother’s life and health, 1035 due to severe and irreversible fetal defects, and zero due to criminal offenses [37]. No information is currently available as regards the number of abortions in the subsequent years. When comparing the number of all live births in Poland (402,000) to the number of terminations (1057) we arrive at 0.26%. Between 1974 and 1991, when liberal abortion laws were in place in Poland, 2,051,164 abortions were performed. The Federation for Women and Family Planning estimates that currently about 80–200 thousand illegal abortions are performed every year in Poland [38]. No possibility of performing a termination in the case of a rape is a burning issue in Poland. It is due to the necessary permission of a prosecutor and the lassitude of the Polish justice which make it impossible to terminate a pregnancy during the first 12 gestational weeks.

### 3.3. Abortion Statistics

Due to the common access to contraception and sex education, developed countries have observed a drop in the number of abortions performed (Table 2). Developing countries still show increasing abortion rates. It is important to note, that in such countries, the procedures are performed by unprofessional medical staff, which leads to a high number of deaths of women undergoing the procedure [6]. No data are available on the number of abortions performed in many countries in Africa, North America, and some countries in Asia. The estimated number of induced miscarriages around the world is shown in Table 2 [39,40]. However, it needs to be considered that the population of people worldwide increased from 5,327,231,061 in 1990 to 7,295,290,765 in 2014 [41].

### 3.4. Sex Education

High-income countries, which usually have a liberal abortion law, are characterized by lower abortion rates [10]. It is due to the common access to contraception and sex education in developed countries. A study conducted in the USA revealed that increased sexuality education within school curricula was associated with lower adolescent birthrates and abortions [42]. However, no statistical significance was demonstrated after comprising sociodemographic characteristics, religion, and abortion laws in individual states. A low level of sex education may lead to an increase in the number of abortions even in countries where contraception is available without prescription [43]. Similarly, an increased frequency of abortions was observed in countries where contraception is not widely used, but terminations are widely available [44]. A study by Stanger–Hall et al. revealed that sex education based on abstinence-only education was ineffective in American pupils. Therefore, it seems justified to provide sex education basing on the presentation of possible contraception methods. The authors suggested including comprehensive sex and STD education into the biology curriculum and a parallel social studies curriculum that addresses risk aversion behaviors and planning for the future [45]. A systematic review concerning sex education demonstrated a higher effectiveness of sex education in the form of digital platforms and blended learning [46]. A study conducted in China including almost 18000 students revealed that individuals who had the basic knowledge concerning sexual health became pregnant and underwent terminations less commonly (OR < 1, *p* < 0.05) [47]. Moreover, preparing medical students to be competent practitioners appears to contribute towards them viewing abortion as an essential aspect of women’s healthcare [48]. A study carried out in Greece showed that despite the possibility of using contraception without prescription it was not frequently used. The authors would like to point out that this may be linked to a low level of sex education at school [49]. Joodaki indicated the significance of sex education also in countries with restrictive abortion laws such as Iran. However, it was suggested that information passed should be adjusted to the law and religion in the country [50].

### 3.5. The Use of Abortion-Inducing Drugs

The ACOG (American College of Obstetrics and Gynecology) recommended the use of abortion-inducing drugs (mifepristone and misoprostol) until 10 gestational weeks [51]. The outbreak of the COVID pandemic contributed to the fact that professional medical personnel became unavailable. Lopez Cabello and Gaitan indicated that the availability of abortion-inducing drugs should be higher in pharmacies and as part of consultations conducted via e-mail [52]. They emphasized the need of self-administration and using telemedicine. The situation necessitated by the pandemic demonstrated that such management is possible and safe. The recommendations were also reflected by those issued by the WHO [53,54]. According to the UK Royal College of Obstetricians and Gynaecologists, in the six weeks following this decision, approximately 16,500 women accessed safe medical abortion at home in England and Wales, at a time when many in-person services were suspended [54,55]. A study conducted in Hungary revealed that out of 59 women who ordered a medical abortion package only 8.5% had a surgical intervention afterwards. All the respondents were satisfied with such a possibility of performing a pregnancy termination [56]. Similar medical abortion effectiveness and safety were achieved in women at 13 or more weeks gestation through Women on Web telemedicine service [57,58]. The self-administration of abortion-inducing drugs and initial advice are also practiced in countries with limited access to health care, such as Tanzania. Such a form of inducing abortion reduces the frequency of complications of abortions performed by individuals not practicing medicine as their profession [59].

## 4. Discussion

Guttmacher Institute, basing on a study conducted in 14 countries worldwide, re-ported that the main reason for abortion was related to the socioeconomic causes and the wish to limit the birth rate in countries characterized by high parity and large populations [60]. It is utopian to think that we can accommodate moral dilemmas, social norms, individual opinions, and the law obligatory in individual countries as regards such a complex issue as abortion. The fundamental task of those responsible for making laws, citizens and medical personnel is to create a real platform enabling the exercising of rights by an individual while respecting one’s own moral norms. The countries where the access to abortion is legally restricted are a challenge [15]. A study conducted by the present authors which compared the number and indications for abortion in Poland and in the UK in the years 2009–2018 showed that the radicalization of abortion law exerted no influence on the decrease in the number of terminations due to fetal indications and that the liberalization of abortion law only promoted an increase in the number of terminations due to social indications [61]. In Russia, abortion law was tightened up in 2016. Vladimir Putin regulated such issues as the access to abortion agents. Therefore, the rate of complications of abortions performed under out-of-hospital conditions is higher in Russia [62].

The Guttmaher Institute analyzed data from a study carried out in 14 countries worldwide and reported that socioeconomic causes and the wish to limit parity were the most common reasons for undergoing [60]. Sex education is undoubtedly a significant issue in terms of reducing the number of terminations due to social indications. The awareness of possible methods of contraception is the best way to prevent unwanted pregnancy which may result in the wish to undergo an abortion. The accessibility to contraception and possibility of using it is another valid issue. Appropriate education at school is half the battle. Therefore, it seems that the most important factors contributing to limiting abortions due to medical indications include an easy access to contraception, awareness, and the possibility of using it. Sex education has been removed from school educational programs in Poland. Therefore, a reasonable alternative may involve the development of websites and the internet for included information concerning sex life, family planning methods, and abortion options verified by sex educators and physicians. The SARS-CoV-2 pandemic contributed to the development of a new tool—telemedicine and increased the frequency of webinar organization which, in the case of sex education, may be available for a larger group of recipients. It is worth mentioning a good example of websites for UK citizens which inform about the possible methods of contraception and the locations where it may be obtained free of charge [63].

Numerous countries accept the self-administration of abortion drugs during the first trimester which excludes the issue of the conscience clause of the medical personnel and facilitates an independent decision concerning abortion [60]. It also allows maintaining anonymity, especially in countries characterized by high stigmatization by the society. Therefore, it seems justified to provide access to OTC abortion-inducing drugs in online shops or at pharmacies [11,64]. Currently, patients from countries with restrictive abortion law, similarly to women living in Poland, have no access to abortion-inducing drugs in the country and order them abroad. Prescriptions are not issued by doctors who are afraid of possible legal consequences. However, they are available online in foreign shops. An increasing number of women use this alternative. The Polish law enforces no punishment on the mother who undergoes an abortion, so Polish women use this legal loophole. Possible legal consequences may only be faced by medical personnel who participate in a termination. However, few women are aware of such a possibility. With regard to the lack of sex education at schools it seems to be justified to develop internet fora and telemedicine to inform patients about possible alternatives.

Currently, in the case of the diagnosis of incurable fetal defects many Polish women travel abroad to undergo pregnancy terminations. Such a situation increases costs and constitutes a logistic challenge for mothers and their families. Numerous organizations are established in Poland to facilitate contact with clinics abroad. The situation contributes to limiting the access to pregnancy termination procedures for women with lower income and poorer access to information concerning alternative solutions available abroad.

### Limitation of the Study

Abortion law undergoes constant and multiple changes, so it is no longer valid at the moment of the publication of this paper. No systematic reviews could be found as regards numerous small countries, especially African ones, so the authors have not comprised all countries worldwide. Moreover, we have no information concerning sex education and the availability of abortion-inducing drugs. The issue of social indications has not been standardized and it is defined in different ways in each country. Similarly, the issue of maternal indications for abortion is defined in a variety of ways. In some countries it includes possible mental complications which may occur after childbirth, while in others it only refers to the direct threat to maternal life.

## 5. Conclusions

The possibility of conscious motherhood at an age appropriate for a woman is a basic right guaranteed to women by the WHO. Some minimal changes should be considered by countries dealing with complications of dangerous abortions and which have restrictive abortion laws:(1)improved sex education and the availability of contraception would prove to be the most beneficial in reducing the percentage of abortions without medical indication;(2)free access to abortion-inducing drugs with adequate information provided by qualified medical personnel in countries with a conscience clause invoked by the personnel;(3)the development of an international network which would facilitate undergoing a pregnancy termination abroad to provide women with access to a legal abortion assisted by professional medical personnel.

## Figures and Tables

**Table 1 healthcare-09-01594-t001:** Countries with restrictive abortion laws with regard to indications for abortion.

Terminations Banned	Terminations Possible Due to the Threat to Maternal Health
Andorra Angola Dominican Republic Congo Egypt El Salvador Gabon Haiti Honduras Iraq Laos Malta Marshall Islands Mauritania Micronesia Nicaragua Palau Philippines Republic of the Congo San Marino Sao Tome and Principe Senegal Suriname Tonga	Afghanistan Antigua and Barbuda Bangladesh Bhutan Brazil Chile Dominica Guatemala Indonesia Iran Ivory Coast Kiribati Lebanon Libya Malawi Mali Mexico Myanmar Nigeria Oman Panama Papua New Guinea Paraguay Poland Solomon Islands Somalia South Sudan Sri Lanka Sudan Syria Tanzania Timor-Leste Tuvalu Uganda United Arab Emirates Venezuela Yemen

**Table 2 healthcare-09-01594-t002:** Estimated number of induced miscarriages (in millions) in selected years around the world.

Region	1990–1994	2003	2008	2010–2014
World (total)	50.4	41.6	43.8	56.3
Africa	4.6	5.6	6.4	8.3
Asia	31.5	25.9	27.3	35.8
Europe	8.2	4.3	4.2	4.4
Latin America	4.4	4.2	4.4	6.5
North America	1.6	1.5	1.4	1.2
Oceania	0.1	0.1	0.1	0.1

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
