# Peer review of "Abortion in Countries with Restrictive Abortion Laws—Possible Directions and Solutions from the Perspective of Poland"

_healthcare, 2021, doi:10.3390/healthcare9111594_

Round 1

Reviewer 1 Report

The authors aim at offering a Polish perspective on the international situation on restrictive abortion laws. This is a very important perspective on which people from all over the world could learn from.

Unfortunately, the authors have not followed the same academic rigour they have shown in their other works and submitted a version that is far too early in its development.

For the following reasons, I would encourage the authors to maintain the aim they promise in the title, but rework a new version of the paper as a new submission:

  • Section 3.1 offers a random overview of a couple of countries without much structure and analytical depth. It would be more interesting to show in what “restrictive laws” consists in and then make a few references to countries that have such laws. This would put at the centre problematic laws (e.g. the level of proof necessary to show sexual assault), on which a Polish perspective can be offered later on.
  • Section 3.2 has interesting information, yet it would have been more helpful to describe more systematically the historical developments that led to the actual situation and then go into details of what is going on in the country
  • The discussion brings in a couple of interesting strategies to address the many problematic issues of the results section, but is not well structured. A reflection from the Polish perspective and experience is often missing. The approach from telemedicine is quite promising and needs to be expanded. Here I found Baum, S. E., Ramirez, A. M., Larrea, S., Filippa, S., Egwuatu, I., Wydrzynska, J., ... & Jelinska, K. (2020). “It's not a seven-headed beast”: abortion experience among women that received support from helplines for medication abortion in restrictive settings. Health Care for Women International, 41(10), 1128-1146. quite useful
  • Table 1 is missing in the document

Generally, the submitted version fails to offer a systematic analysis of the current situation and tends to merely depict a series of important facts. A new version needs to offer a more detailed analysis and allow the reader to benefit from the authors expertise.

Some minor issues:

  • Ln 27 “giving women the right” it is more about recognizing the right
  • The conscience clause or conscientious objection clause is interpreted very differently in each country, see for instance the ongoing special issue in Developing World Bioethics
  • Ln 69 “eugenic reasons” are mostly prohibited, while it is true that laws can have “eugenic effects”
  • Ln 83-84 “but some women still treat abortion as a form of emergency contraception” this needs a reference or needs to reformulated, as reasonable people generally do not opt for a chirurgical intervention when less invasive methods are at their disposal

Author Response

Dear Reviewer,

At the beginning, we want to thank you for all valuable remarks.  The manuscript has been revised and reorganised according to your suggestions. Detailed answer is provided below.

               Section 3.1 offers a random overview of a couple of countries without much structure and analytical depth. It would be more interesting to show in what “restrictive laws” consists in and then make a few references to countries that have such laws. This would put at the centre problematic laws (e.g. the level of proof necessary to show sexual assault), on which a Polish perspective can be offered later on.

We have changed the section according to your suggestions.

                Section 3.2 has interesting information, yet it would have been more helpful to describe more systematically the historical developments that led to the actual situation and then go into details of what is going on in the country

We have changed the section according to your suggestions.

                The discussion brings in a couple of interesting strategies to address the many problematic issues of the results section, but is not well structured. A reflection from the Polish perspective and experience is often missing. The approach from telemedicine is quite promising and needs to be expanded. Here I found Baum, S. E., Ramirez, A. M., Larrea, S., Filippa, S., Egwuatu, I., Wydrzynska, J., ... & Jelinska, K. (2020). “It's not a seven-headed beast”: abortion experience among women that received support from helplines for medication abortion in restrictive settings. Health Care for Women International, 41(10), 1128-1146. quite useful

We have changed the section according to your suggestions.

Table 1  missing in the document

The Table 1 had been added and changed on Table 2

Generally, the submitted version fails to offer a systematic analysis of the current situation and tends to merely depict a series of important facts. A new version needs to offer a more detailed analysis and allow the reader to benefit from the authors expertise.

We have changed the article according to your suggestions.

 Ln 27 “giving women the right” it is more about recognizing the right

We have changed the sentence according to your suggestions.

               The conscience clause or conscientious objection clause is interpreted very differently in each country, see for instance the ongoing special issue in Developing World Bioethics

We have changed the article according to your suggestions.

                Ln 69 “eugenic reasons” are mostly prohibited, while it is true that laws can have “eugenic effects”

We have changed the sentence according to your suggestions.

               Ln 83-84 “but some women still treat abortion as a form of emergency contraception” this needs a reference or needs to reformulated, as reasonable people generally do not opt for a chirurgical intervention when less invasive methods are at their disposal

Reference has been added and the sentence was refolmuled in other section.

Best regards,

Kornelia Zareba and Co-authors

Reviewer 2 Report

We could know that abortion is an important issue for Poland by reading this paper. However, some pointed need to be clarifies or revised.

  1.  (Introduction) Please explain more about a novelty of this study. I think there are some review or ecological studies investigating regional differences in abortion laws in the world. You should cite those studies and explain what is revealed in this study for the first time.
  2.  (Results, lines 84-86) I think these sentences should be written in Discussion, but not in Results.
  3. (Results) Why did you summarize only legal aspects of abortions across countries? In lines 33-36 in Introduction, it is written that the aim of review is to determine~, but you mainly reviewed legal aspect of abortion. It might be better to crease a Table showing abortion law, sex education, and abortion drug use by country to summarize findings of this study. Or, it might be better to write about reviews of sex education and abortion drugs use too in the results.
  4. (Results. line 165) Where is Table 1 ?
  5. (Discussion, lines 178-205) I think these are a part of the results of this study.
  6. (Discussion) Please write limitations in Discussion. No limitations exist in this study?
  7. (Conclusions) How the sentence 1) was derived from the results of this study? Can you affirm that these are the most beneficial in reducing abortions? In addition, please write about regional differences in sex education or effect of sex education in Results if you want to write this statement in conclusion.

Author Response

Dear Reviewer,

At the beginning, we want to thank you for all valuable remarks.  The manuscript has been revised according to your suggestions. Detailed answer is provided below.

     (Introduction) Please explain more about a novelty of this study. I think there are some review or ecological studies investigating regional differences in abortion laws in the world. You should cite those studies and explain what is revealed in this study for the first time.

We have added the information according to your suggestions.

                (Results, lines 84-86) I think these sentences should be written in Discussion, but not in Results.

We have changed the article according to your suggestions.

               (Results) Why did you summarize only legal aspects of abortions across countries? In lines 33-36 in Introduction, it is written that the aim of review is to determine~, but you mainly reviewed legal aspect of abortion. It might be better to crease a Table showing abortion law, sex education, and abortion drug use by country to summarize findings of this study. Or, it might be better to write about reviews of sex education and abortion drugs use too in the results.

The reviews about sex education and abortion had been added, however we don’t have anty informations about it in many countries.

               (Results. line 165) Where is Table 1 ?

The Table had been added and changed on Table 2

              (Discussion, lines 178-205) I think these are a part of the results of this study.

We have changed the article according to your suggestions.

                (Discussion) Please write limitations in Discussion. No limitations exist in this study?

Limitations have been added

                (Conclusions) How the sentence 1) was derived from the results of this study? Can you affirm that these are the most beneficial in reducing abortions? In addition, please write about regional differences in sex education or effect of sex education in Results if you want to write this statement in conclusion.

We have changed the article according to your suggestions.

Best regards,

Kornelia Zareba and Co-authors

Round 2

Reviewer 1 Report

The authors have made major improvements towards improving the manuscript. As I originally recommended to reject the manuscript, there are still a few points I think need improvement:

p1 In the first paragraph, radicalization/radical needs to be discussed in which direction

p1. Ln41 “many countries” is too vague, a rough indicator, e.g. a quarter, would be better, if no precise number can be found

p.2 ln 54 “currently”, at the time of this writing, November 2021 , might be better for later readers

p2. Ln 65-67 it is unclear what the “correspondence argument” is

p.3 ln 100 “smaller Asian islands” does not match the list (e.g. Kiribati is in Polynesia and Indonesia is rather big)

p.4 ln 147-160 The information on China is too general and out of date, please double-check the current legislation on sex-selection of India, it might also be out of date.

p.5 Table 2 somewhere a reference to population growth needs to be included, if we take that in consideration we have a substantial declination of the rate of abortion by population size

p.6 ln 253 it is unclear what “limit parity” means in this context

p.7 ln 267-278 the Polish perspective suggested in the title is missing for the sex education part.

p.7 ln 284 “authorizing the actual state” is somewhat unclear

p.7 ln 295-298 it would be good to have 1-2 sentences as a critical reflection on the injustice of de facto unequal access to abortion for poorer women

p.7 ln 302-304 “no information is available” is not accurate as laws are generally published. No systematic reviews or studies could be found with our methodology, or something similar, is more likely.

p.8  “Conclusions” free access without information does not lead to more use. The role of adequate information needs to be specified, as it is central (and also discussed in the paper).

Author Response

Dear Reviewer,

At the beginning, we want to thank you for all valuable remarks. We are happy that you enjoyed this topic. The manuscript has been revised according to your suggestions. Detailed answer is provided below and highlighted in the article in green.

p1 In the first paragraph, radicalization/radical needs to be discussed in which direction

We have added the information according to your suggestions.

p1. Ln41 “many countries” is too vague, a rough indicator, e.g. a quarter, would be better, if no precise number can be found

We have added the numbers according to your suggestions.

p.2 ln 54 “currently”, at the time of this writing, November 2021 , might be better for later readers

We have changed the sentence according to your suggestions.

p2. Ln 65-67 it is unclear what the “correspondence argument” is

We have added the definition according to your suggestions.

p.3 ln 100 “smaller Asian islands” does not match the list (e.g. Kiribati is in Polynesia and Indonesia is rather big)

We have changed the sentence according to your suggestions using „Asian islands and Oceania”.

p.4 ln 147-160 The information on China is too general and out of date, please double-check the current legislation on sex-selection of India, it might also be out of date.

We have changed the information about China according to your suggestions. The legislation on sex-selection in India is still current.

p.5 Table 2 somewhere a reference to population growth needs to be included, if we take that in consideration we have a substantial declination of the rate of abortion by population size

We have added the information according to your suggestions.

p.6 ln 253 it is unclear what “limit parity” means in this context

We have changed the sentence according to your suggestions

p.7 ln 267-278 the Polish perspective suggested in the title is missing for the sex education part.

We have added the information according to your suggestions.

p.7 ln 284 “authorizing the actual state” is somewhat unclear

We have changed the sentence according to your suggestions.

p.7 ln 295-298 it would be good to have 1-2 sentences as a critical reflection on the injustice of de facto unequal access to abortion for poorer women.

We have added the information according to your suggestions.

p.7 ln 302-304 “no information is available” is not accurate as laws are generally published. No systematic reviews or studies could be found with our methodology, or something similar, is more likely.

We have changed the sentence according to your suggestions.

p.8  “Conclusions” free access without information does not lead to more use. The role of adequate information needs to be specified, as it is central (and also discussed in the paper).

We have added the information according to your suggestions.

Best regards,

Kornelia Zareba and Co-authors

Reviewer 2 Report

  1.     (Abstract and Conclusions) Please write about results or findings of this study in Abstract.
  2.  (Results, Table 2) How did you calculate the estimated numbers ? In addition, line numbers are included in Table2.
  3.  (Discussion, line 267) adta →data
  4. (Supplementary materials line 322) Table 1 (2?) should be placed out of the manuscript if it is supplementary material. However, it may need to be placed in the manuscript rather than supplementary material.

Author Response

Dear Reviewer,

At the beginning, we want to thank you for all valuable remarks. We are happy that you enjoyed this topic. The manuscript has been revised according to your suggestions. Detailed answer is provided below and highlighted in the article in green.

               (Abstract and Conclusions) Please write about results or findings of this study in Abstract.

We have added the world according to your suggestions.

 (Results, Table 2) How did you calculate the estimated numbers ? In addition, line numbers are included in Table2.

We have added the Bibliography.

              (Discussion, line 267) adta →data

We have changed the world according to your suggestions.

(Supplementary materials line 322) Table 1 (2?) should be placed out of the manuscript if it is supplementary material. However, it may need to be placed in the manuscript rather than supplementary material.

The Tables have been placed in the text. The information about Supplementary materials have been deleted.

Best regards,

Kornelia Zareba and Co-authors